# Integrated extracellular microRNA profiling for ovarian cancer screening

Akira Yokoi[1,2], Juntaro Matsuzaki [1], Yusuke Yamamoto[1], Yutaka Yoneoka[3], Kenta Takahashi[3], Hanako Shimizu[3], Takashi Uehara[3], Mitsuya Ishikawa[3], Shun-ichi Ikeda[3], Takumi Sonoda[1], Junpei Kawauchi[4], Satoko Takizawa[4], Yoshiaki Aoki[5], Shumpei Niida[6], Hiromi Sakamoto[7], Ken Kato [8], Tomoyasu Kato [3] & Takahiro Ochiya[1]

A major obstacle to improving prognoses in ovarian cancer is the lack of effective screening methods for early detection. Circulating microRNAs (miRNAs) have been recognized as promising biomarkers that could lead to clinical applications. Here, to develop an optimal detection method, we use microarrays to obtain comprehensive miRNA profiles from 4046 serum samples, including 428 patients with ovarian tumors. A diagnostic model based on expression levels of ten miRNAs is constructed in the discovery set. Validation in an independent cohort reveals that the model is very accurate (sensitivity, 0.99; specificity, 1.00), and the diagnostic accuracy is maintained even in early-stage ovarian cancers. Furthermore, we construct two additional models, each using 9–10 serum miRNAs, aimed at discriminating ovarian cancers from the other types of solid tumors or benign ovarian tumors. Our findings provide robust evidence that the serum miRNA profile represents a promising diagnostic biomarker for ovarian cancer.

[1] Division of Molecular and Cellular Medicine, National Cancer Center Research Institute, 05-01-01 Tsukiji, Chuo-ku, Tokyo 104-0045, Japan. [2] Department of Obstetrics and Gynecology, Nagoya University Graduate School of Medicine, 65 Tsuruma-cho, Showa-ku, Nagoya 466-8550, Japan. [3] Department of Gynecology, National Cancer Center Hospital, 5-1-1 Tsukiji, Chuo-ku, Tokyo 104-0045, Japan. [4] New Frontiers Research Institute, Toray Industries, 6-10-1 Tebiro, Kamakura city, Kanagawa 248-0036, Japan. [5] Division of Bioinformatics, Dynacom Co., Ltd., World Business Garden E25, 2-6-1 Nakase, Mihama-ku, Chiba city, Chiba 261-7125, Japan. [6] Medical Genome Center, National Center for Geriatrics and Gerontology, 7-430 Morioka-cho, Obu, Aichi 474-8511, Japan. [7] Department of Clinical Genomics, Fundamental Innovative Oncology Core, National Cancer Center Research Institute, Tokyo 104-0045, Japan. [8] Department of Gastrointestinal Medical Oncology, National Cancer Center Hospital, 5-1-1 Tsukiji, Chuo-ku, Tokyo 104-0045, Japan. These authors contributed equally: Akira Yokoi, Juntaro Matsuzaki.  Correspondence and requests for materials should be addressed to T.O. (email: tochiya@ncc.go.jp)

Ovarian cancer is the gynecologic malignancy with the highest death rate[1], and the number of cases is increasing worldwide[2]. The high degree of lethality is largely a reflection of the fact that the disease is often detected late in its progression: approximately 75% of patients present at stage III or IV, as defined by the International Federation of Gynecology and Obstetrics (FIGO), with widespread metastasis in the peritoneal cavity[3]. The 5-year survival rate of stage I patients is 90%, whereas that of patients in stage III–IV is less than 10%[4]. Unfortunately, no effective screening method for this cancer yet exists. Pelvic examination and trans-vaginal ultrasound are the standard methods for diagnosing ovarian tumors, but are inadequate for screening because they lack sensitivity and cause stress to the patient[5]. CA125 is a widely accepted serum biomarker protein for ovarian cancer[6], but is a poor indicator of early-stage patients, with a sensitivity of approximately 40%[7,8]. Because the ovaries are completely intraperitoneal organs, it is currently impossible to diagnose ovarian cancer without surgical resection. However, needle biopsies for early-stage ovarian tumors should be avoided because cancer cells easily disseminate into the peritoneal cavity, and puncture can promote peritoneal metastasis[9]. Consequently, less-invasive biomarkers for ovarian cancer that could achieve early detection and monitor the course of the disease, potentially leading to cancer screening tests, are urgently needed.

Extracellular RNA (exRNA), including circulating microRNA (miRNA), has recently received a great deal of research attention. MiRNAs, small non-coding RNAs 20–25 nucleotides in length, regulate gene expression in cells by repressing the translation of their target genes or degrading their target mRNAs[10]. miRNAs secreted from cells exist stably in body fluids within extracellular vesicles (EVs), including exosomes, or bound to proteins or lipids[11], and play crucial roles in intercellular communication[12]. Recent work revealed that circulating miRNAs reflect physiological and pathological status, and are thus promising biomarkers for various disease states[13,14].

Although several reports demonstrate the suitability of circulating miRNAs as cancer biomarkers[15,16], these molecules are still considered insufficient for clinical applications, primarily due to the lack of large-scale validation and inconsistencies among detection devices[17]. To standardize platforms for collection and detection of serum miRNAs, we recently launched a national project in Japan, entitled Development and Diagnostic Technology for Detection of miRNA in Body Fluids. This project includes the comprehensive characterization of serum miRNA profiles of 13 types of human cancers, including ovarian cancer, in more than 40,000 patients, using the same platform and technology.

Here, we describe our identification of promising biomarkers for the diagnosis of ovarian cancer using serum samples obtained from 4046 women, including 428 patients with ovarian tumors. Our primary aim is to develop a novel screening strategy capable of discriminating cancer patients from healthy women. In addition, comprehensive profiles of circulating miRNAs, which were obtained from all samples, enabled us to generate optimal diagnostic models for ovarian cancer.

## Results

**Assay design.** A total of 4052 serum samples were analyzed by miRNA microarray, yielding comprehensive miRNA expression profiles. After exclusion of six samples with low-quality results, 4046 samples remained for analysis, including 333 ovarian cancers, 66 borderline ovarian tumors, 29 benign ovarian tumors, 2759 non-cancer controls, and 859 other solid cancers. Using these samples, we constructed three kinds of discrimination models: (1) ovarian cancer vs. non-cancer, (2) ovarian cancer vs. the other cancers + non-cancer, and (3) ovarian cancer vs. borderline/benign ovarian tumors + non-cancer.

**Selection of circulating miRNA biomarker candidates.** To focus on extracellular miRNAs released from ovarian cancer cells, we evaluated miRNA expression in EVs, including exosomes, from 12 ovarian cancer cell lines (listed in Supplementary Table 1). A total of 858 miRNAs detected in exosomes derived from at least one cell line were considered as candidate ovarian cancer-released miRNAs. We proceeded to further select miRNA candidates using a human serum dataset. Specifically, miRNAs with a signal value $>2^6$ in more than 50% of samples were selected as robust biomarkers in serum samples from ovarian cancers. Based on this analysis, 648 miRNAs (of the previously selected 858) were excluded due to low signal. Ultimately, 210 miRNAs were selected for further analyses (Supplementary Figure 1).

**Identifying the best combination of miRNAs for screening.** To develop models for discrimination between ovarian cancer and non-cancer samples, we randomly divided 320 ovarian carcinoma samples and 2759 non-cancer samples into two groups: the discovery set and validation set (Fig. 1a). Samples of non-epithelial ovarian cancers, borderline ovarian tumors, and benign ovarian tumors were allocated to the validation set to evaluate whether the model for detecting ovarian carcinomas would also detect non-epithelial ovarian cancers, borderline tumors, or benign ovarian tumors. Participant characteristics are described in Table 1.

First, we identified the best 10 miRNAs with the highest AUC values, termed pivot miRNAs, as shown in Supplementary Table 2. A predictive model was created based on these pivot miRNAs, and other miRNAs were used to compensate for their diagnostic performance. Thus, the levels of the pivot miRNAs are the most important factors governing model performance, and must be reproducible on independent platforms if these models are to proceed to clinical application. Accordingly, we investigated the expression of pivot miRNAs by qRT-PCR. Around 60% of miRNAs were detectable in the quantitative performance assay and were used for subsequent analysis (Supplementary Figure 2); an R-value of 0.9 which was greater than 0.9 was considered as a cut-off value for inclusion. Then, to assess reproducibility of miRNAs between microarray and qRT-PCR analyses, the levels of each miRNA were plotted as histograms using ten patient samples randomly selected from the non-cancer A ($N = 5$) and ovarian cancer cohorts ($N = 5$), as shown in Supplementary Figure 3. If the R-values were less than zero, we considered the miRNA to be validated by qRT-PCR. The prediction models were further developed using these pivot miRNAs.

Using Fisher's linear discriminant analysis, we designed comprehensive discriminants consisting of one to ten miRNAs from the discovery set (Supplementary Table 3). Based on the optimal level of accuracy, the analysis identified a combination of ten miRNAs (miR-320a, miR-665, miR-3184-5p, miR-6717-5p, miR-4459, miR-6076, miR-3195, miR-1275, miR-3185, and miR-4640-5p) that provided the best discrimination in the discovery set [diagnostic index = $(0.581) \times$ miR-320a + $(0.691) \times$ miR-665 + $(-0.704) \times$ miR-3184-5p + $(-0.313) \times$ miR-6717-5p + $(-1.302) \times$ miR-4459 + $(0.729) \times$ miR-6076 + $(0.676) \times$ miR-3195 + $(0.716) \times$ miR-1275 + $(0.672) \times$ miR-3185 + $(-0.384) \times$ miR-4640-5p—9.375 [model 1]; area under curve (AUC): 1.00; sensitivity: 1.00; specificity: 1.00]. Some single miRNAs were also statistically effective in distinguishing cancer patients (Fig. 1b, c). The diagnostic performance of model 1 was confirmed in the validation set, revealing that the model was very accurate (AUC: 1.00; sensitivity: 0.99; specificity: 1.00) (Fig. 1c). Although model

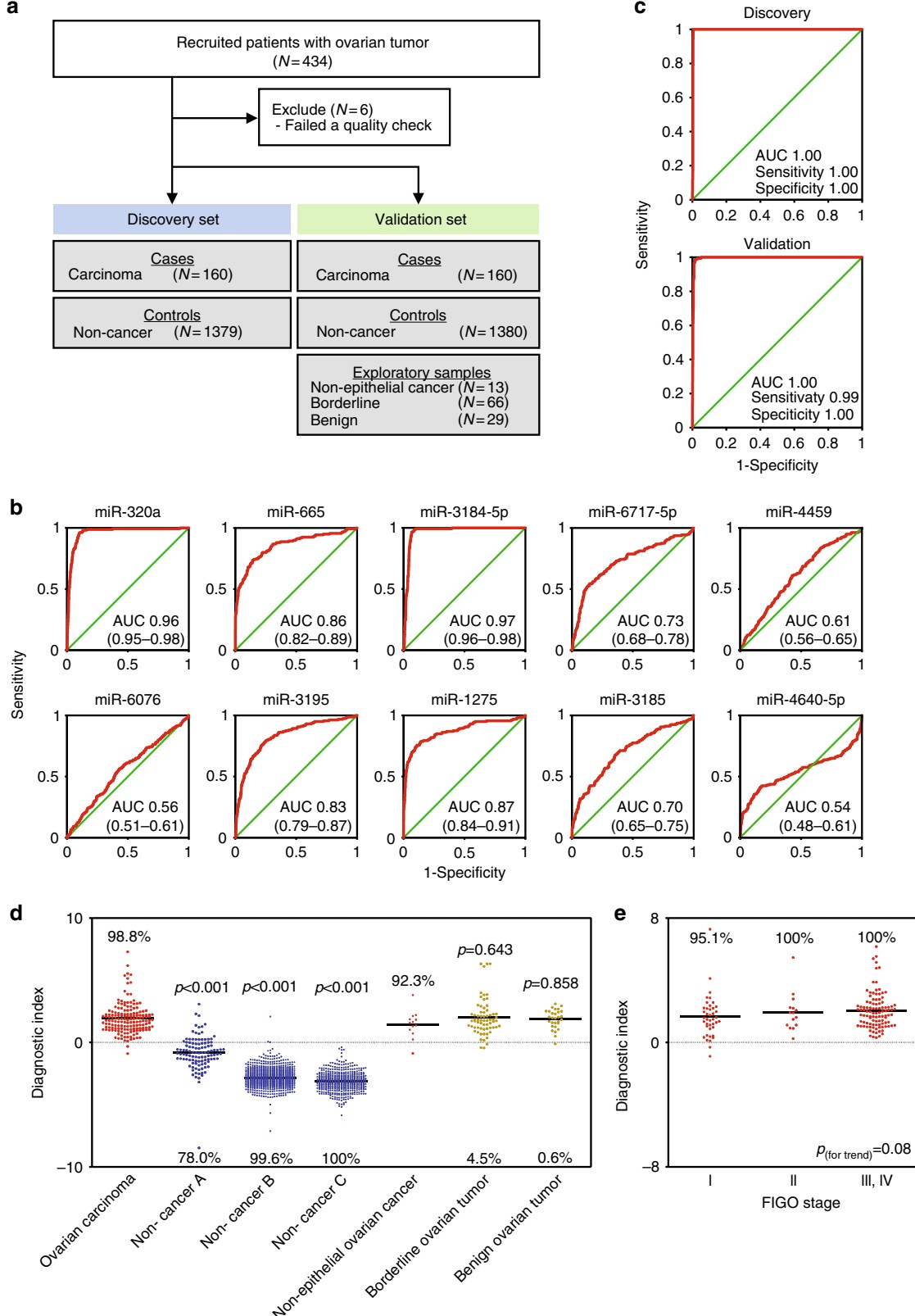

1 successfully discriminated non-epithelial ovarian cancer patients from non-cancer controls, it could not distinguish ovarian cancer patients from those with borderline and benign tumors (Fig. 1d).

According to the FIGO criteria, we subdivided ovarian cancers based on the stage: stage I, stage II, and stage III–IV. Although the serum samples were collected before any type of treatment, FIGO stage was diagnosed using tissue specimens collected during surgery. The stage III and IV groups were combined because most patients diagnosed at advanced stages underwent chemotherapy prior to surgery, and consequently we were unable to correctly categorize them individually as stage III or IV. As shown in

**Fig. 1** Development of the ovarian cancer screening model (model 1). **a** Work flow of patients for developing prediction model 1. Serum samples were obtained from 3007 subjects, including 428 patients with ovarian tumors and 2759 non-cancer donors. The sample set was divided into two groups, the discovery set and validation set. **b** ROC curves for detecting cancer patients using a combination of ten miRNAs selected for prediction model 1. **c** Diagnostic performance of the ten selected miRNAs in the discovery set and validation set. **d** Diagnostic index using prediction model 1 in the validation set (ovarian cancer, 160; non-cancer A, 109; non-cancer B, 789; non-cancer C, 482; non-epithelial ovarian cancer, 13; borderline ovarian tumor, 66; and benign ovarian tumor, 29). Each diagnostic accuracy (%) is included. *p* values were calculated by $\chi^2$ test. **e** Diagnostic index for each FIGO stage using prediction model 1. Each diagnostic accuracy (%) is indicated. *N* = stage I, 82; stage II, 33; and stage III–IV, 218. The *p* value was calculated using Pearson's correlation analysis

**Table 1 Participant characteristics in model 1**

| Characteristics | Total ($N = 3187$) | Discovery set ($N = 1539$) | | | Validation set ($N = 1648$) | | | |
|---|---|---|---|---|---|---|---|---|
| | *N* | *N* | Mean | SD | *N* | Mean | SD | *p* |
| Ovarian carcinoma | 333 | 160 | | | 160 | | | |
| Age, years | | | 56.8 | 11.5 | | 57.1 | 11.6 | 0.97[a] |
| Histopathological subtypes | | | | | | | | 0.29[b] |
| Serous | 182 | 90 | | | 92 | | | |
| Clear cell | 64 | 37 | | | 27 | | | |
| Endometrioid | 43 | 17 | | | 26 | | | |
| Mucinous | 14 | 6 | | | 8 | | | |
| Other epithelial carcinoma | 17 | 10 | | | 7 | | | |
| Non-epithelial carcinoma | 13 | | | | 13 | | | |
| Stage | | | | | | | | 0.97[b] |
| I | 82 | 39 | | | 43 | | | |
| II | 33 | 15 | | | 18 | | | |
| III–IV | 218 | 106 | | | 112 | | | |
| Borderline ovarian tumor | 66 | | | | 66 | | | |
| Benign ovarian tumor | 29 | | | | 29 | | | |
| Non-cancer cohort | 2759 | 1379 | | | 1380 | | | |
| Institute A | 209 | 100 | | | 109 | | | |
| Institute B | 1581 | 792 | | | 789 | | | |
| Institute C | 969 | 487 | | | 482 | | | |

[a]Student's *t*-test
[b]$\chi^2$ test

Fig. 1e, advanced stages had higher diagnostic indices. Notably, model 1 classified 95.1% of stage I patients as positive, indicating that this model is highly suitable for early detection. Thus, this combination of ten miRNAs represents a promising biomarker for ovarian cancer screening.

**Further potentials for ovarian cancer biomarkers**. To investigate whether the serum miRNA profile can distinguish ovarian cancers from other solid cancers, we developed another model. For this purpose, we also comprehensively analyzed female serum miRNA profiles of breast carcinoma ($N = 115$), pancreatic ductal adenocarcinoma ($N = 115$), colorectal adenocarcinoma ($N = 115$), hepatocellular carcinoma ($N = 81$), esophageal squamous cell carcinoma ($N = 88$), gastric adenocarcinoma ($N = 115$), lung carcinoma ($N = 115$), and bone and soft tissue sarcoma ($N = 115$). As for model 1, 320 ovarian carcinoma samples were randomly divided into a discovery set and validation set (Fig. 2a). Fifteen samples of each non-ovarian cancer and non-cancer controls were randomly selected and allocated to the discovery set. The other non-ovarian cancer samples, 100 non-cancer samples, and samples of non-epithelial ovarian cancers, borderline ovarian tumors, and benign ovarian tumors were allocated to the validation set. Participant characteristics are described in Table 2.

This model was developed in the same manner as model 1: we first identified pivot miRNAs (Supplementary Table 4) and then validated the miRNAs by qRT-PCR (Supplementary Figure 2 and 3). Using the validated pivot miRNAs, we performed Fisher's

linear discriminant analysis, this time identifying a different combination of ten miRNAs (miR-4687-3p, miR-939-5p, miR-5739, miR-211-3p, miR-1273g-3p, miR-3663-3p, miR-4726-5p, miR-4745-5p, miR-1268b, and miR-658) that provided the best discrimination within the discovery set [diagnostic index = $(0.996) \times$ miR-4687-3p + $(-0.741) \times$ miR-939-5p + $(0.718) \times$ miR-5739 + $(-0.798) \times$ miR-211-3p + $(0.719) \times$ miR-1273g-3p + $(1.036) \times$ miR-3663-3p + $(0.520) \times$ miR-4726-5p + $(-0.583) \times$ miR-4745-5p + $(0.786) \times$ miR-1268b + $(-0.223) \times$ miR-658 − 25.0 (model 2); AUC: 0.97; sensitivity: 0.94; specificity: 0.99], although none of the individual miRNAs was sufficient to discriminate cancer patients from healthy subjects (Supplementary Table 5, Fig. 2b, c). The diagnostic performance of model 2 was confirmed in the validation set (AUC: 0.87; sensitivity: 0.84; specificity: 0.90) (Fig. 2c). Although model 2 misdiagnosed around half of sarcoma and esophageal cancer samples as ovarian cancer, it adequately distinguished ovarian cancer patients from the other cancer types (Fig. 2d).

However, neither model 1 nor model 2 distinguished ovarian cancer patients from patients with borderline and benign tumors. To this end, we performed cluster analysis and principal component analysis (PCA) between cancer and benign tumors, revealing relatively distinct miRNA profiles between them (Fig. 3a, b). These data prompted us to establish another model to discriminate these two groups (i.e., cancer and benign tumors). The third model was developed in the same manner as described above (Table 2, Fig. 3c, Supplementary Table 6, Supplementary Figure 2 and 3), and we identified a combination of nine miRNAs (miR-663b, miR-4730, miR-642a-3p, miR-658, miR-486-3p,

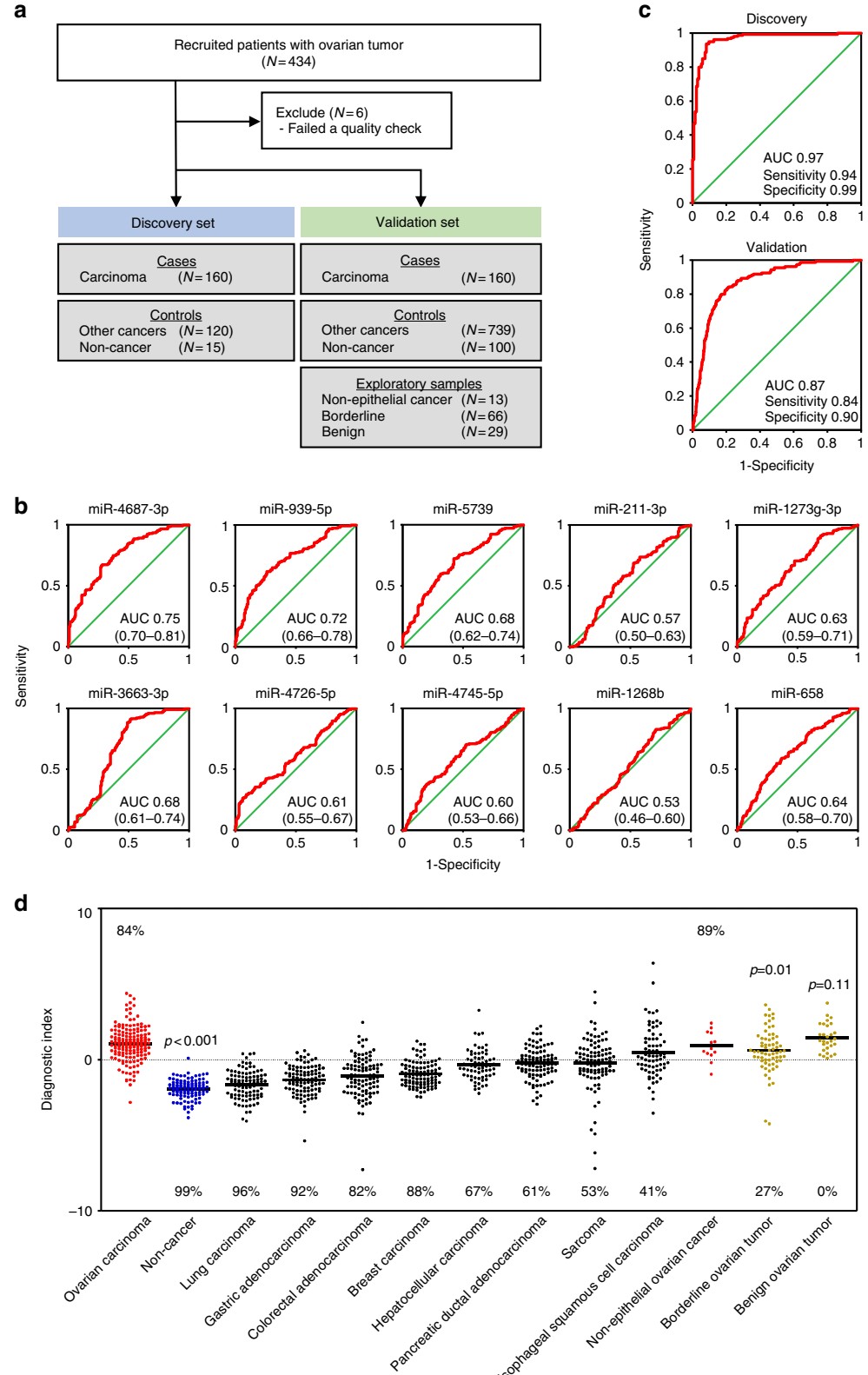

miR-1246, miR-1207-5p, miR-4419b, and miR-6124) that provided the best discrimination in both the discovery set [diagnostic index = (0.715) × miR-663b + (−0.710) × miR-4730 + (0.254) × miR-642a-3p + (0.628) × miR-658 + (0.013) × miR-486-3p + (−0.0519) × miR-1246 + (0.317) × miR-1207-5p + (0.179) × miR-4419b + (−0.264) × miR-6124 − 7.2 (model 3); AUC: 0.72; sensitivity: 0.64; specificity: 0.80] and the validation set (AUC: 0.86; sensitivity: 0.82; specificity: 0.91) (Supplementary Table 7 and Fig. 3d, e). Consistent with its diagnostic performance in the validation set, model 3 distinguished ovarian cancer from non-cancer controls but could not efficiently discriminate patients with benign or borderline tumors from ovarian cancer patients (Fig. 3b, f and Supplementary Figure 4).

**Fig. 2** Development of the ovarian cancer detection model (model 2). **a** Work flow of patients for development of prediction model 2. Serum samples were obtained from 1402 subjects, including 428 patients with ovarian tumors, 859 with other cancers, and 115 non-cancer controls (from non-cancer control B). The sample set was divided into two groups, the discovery set and validation set. **b** ROC curves for detecting cancer patients using the miRNAs selected for prediction model 2. **c** Diagnostic performance of the ten selected miRNAs in the discovery set and validation set. **d** Diagnostic index using prediction model 2 in the validation set (ovarian carcinoma, 160; breast carcinoma, 100; colorectal adenocarcinoma, 100; esophageal squamous cell carcinoma, 73; gastric adenocarcinoma, 100; hepatocellular carcinoma, 66; lung carcinoma, 100; pancreatic ductal adenocarcinoma, 100; sarcoma, 100; non-cancer, 100; non-epithelial ovarian cancer, 13; borderline ovarian tumor, 66; and benign ovarian tumor, 29). Each diagnostic accuracy (%) is indicated. The p values were calculated by $\chi^2$ test

**Characterizing the performance of prediction models**. To further assess the models developed herein, we investigated the influence of patient background. Patients at advanced FIGO stages had higher diagnostic indices in all models (Fig. 1e and Supplementary Figure 5). Because participant age was not matched between ovarian cancer samples and non-cancer controls, we performed age-adjusted logistic regression analysis. As shown in Table 3, all three models could predict the presence of ovarian cancer with statistical significance after adjustment for age.

We also investigated the relationship of histopathological subtypes in ovarian cancer. For this purpose, we categorized ovarian carcinoma samples into four major histological subtypes with distinct molecular and pathological characteristics: serous ($N = 182$), clear-cell ($N = 64$), endometrioid ($N = 43$), and mucinous ($N = 14$). Unclassifiable ovarian carcinoma samples were excluded from this analysis. PCA mapping suggested that miRNA profiles did not differ significantly among the four subtypes (Supplementary Figure 6a). By contrast, hierarchical cluster analysis showed that serum miRNA profiles of patients with ovarian carcinoma could be categorized into three clusters (Supplementary Figure 6b). When we compared age, histopathological subtypes, and disease stage among the three clusters, however, we detected no significant differences (Supplementary Table 8). This suggests that histopathological subtype does not affect the serum miRNA profile of ovarian carcinoma. In fact, the diagnostic sensitivities of the three models exhibited no obvious differences among the histopathological subtypes (Supplementary Figure 7).

**Discussion**

In this study, we analyzed serum miRNAs in a large sample set ($N = 4046$), including 428 patients with ovarian tumors. From all samples, we obtained comprehensive profiles of 2588 miRNAs using highly sensitive miRNA microarray analysis on a standardized platform (3D-Gene®, Toray Industries, Inc., Tokyo, Japan)[18]. Previously, the largest study aimed at evaluating the diagnostic performance of circulating miRNAs in ovarian cancer included 360 patients[19]. In that study, however, the authors pooled serum samples of ten early-stage cases (stage I), ten late-stage cases (stage IIIc–IV), and ten healthy controls, and analyzed these three pool samples using a TaqMan low-density array (667 miRNAs) in the initial step of their analyses[20]. They then selected candidate miRNAs, but evaluated only a limited number of miRNAs by quantitative reverse transcription PCR (qRT-PCR). Therefore, the present study is the first large-scale comprehensive analysis of circulating miRNAs in ovarian cancer. We also analyzed exosomal miRNAs using 12 ovarian cancer cell lines, and selected as marker candidates miRNAs that can be encapsulated in exosomes and released from ovarian cancer cells. Using such large-scale data, we confirmed previous suggestions that ovarian cancer patients can be accurately discriminated from non-cancer controls using serum miRNA profiles (model 1). In the current study, we did not isolate exosomes from serum due to limitations on sample volume; moreover, such isolation is not compatible

with high throughput. However, it is possible that miRNAs are packaged in exosomes and that they might play functional roles.

Before developing each prediction model, we performed qRT-PCR validation of pivot miRNAs to maximize the usefulness of models for further clinical applications. Several miRNAs were not well validated. miRNA microarray can detect iso-miRNAs, but qRT-PCR requires perfect consistency with the primer sequences[21]. This may have been responsible for the validation failure. However, this validation step is important for ensuring and improving the value of models.

Previous studies did not investigate whether the circulating miRNA profile of ovarian cancer patients is distinct from those of other solid cancers. Although the profiles of well-studied circulating miRNA biomarker candidates, such as miR-21, miR-221, and miR-155, are altered in ovarian cancer patients[19], it is not possible to distinguish ovarian cancers using these miRNAs because their levels are also altered in patients with other cancers. Our data revealed that ovarian cancer patients could be sufficiently discriminated from those with lung, gastric, breast, hepatic, colorectal, and pancreatic carcinoma, but not from those with sarcoma and esophageal squamous cell carcinoma (model 2). Although we do not currently have evidence of similarities among those cancers, future investigations could elucidate a connection. This information about analyses across various cancer types will also be useful in the clinical application of serum miRNA panels for the monitoring of ovarian cancer.

In addition, we investigated whether the serum miRNA profile could discriminate ovarian cancers from borderline or benign ovarian tumors. Distinguishing benign tumors from malignant cancers is a major concern for gynecologists, and a less-invasive diagnostic method would be of great clinical value. Our model 3 could not discriminate ovarian cancer from benign tumors (Fig. 3f). Several previous reports using much smaller sample sets suggested that it would be possible to discriminate malignant from benign ovarian tumors using circulating miRNAs[22–24]. However, none of those studies reported diagnostic performance in terms of sensitivity and specificity, and no previous study has developed a model capable of discriminating between malignant and borderline ovarian tumors. In this study, we also found it difficult to discriminate between ovarian cancer and borderline ovarian tumors. However, because the sample sizes of borderline or benign ovarian tumors were insufficient for model construction, further large-scale validation studies are needed to resolve this issue.

In 2012, The Cancer Genome Atlas (TCGA) Research Network released the whole genome profiles of ovarian cancer, including miRNA profiles[25]. TCGA analyzed 489 ovarian cancer tissues, and performed miRNA microarray analyses on all samples. The results identified three miRNA profile subtypes associated with different survival outcomes. The serum miRNA profiles of ovarian carcinoma patients in this study were also categorized into three groups (Supplementary Figure 6b). Because the details of miRNA clustering were not described in TCGA[25], it was difficult to compare the clustered patterns of miRNAs between sera and tissues. We showed that serum miRNA clusters were not

**Table 2 Participant characteristics in models 2 and 3**

| Characteristics | Total (N = 1402) | Model 2 Discovery set (N = 295) | | | Model 2 Validation set (N = 1107) | | | | Model 3 Discovery set (N = 223) | | | Model 3 Validation set (N = 320) | | | |
|---|---|---|---|---|---|---|---|---|---|---|---|---|---|---|---|
| | N | N | Mean | SD | N | Mean | SD | p | N | Mean | SD | N | Mean | SD | p |
| Ovarian carcinoma | 333 | 160 | | | 160 | | | | 160 | | | 173 | | | |
| Age, years | | | 56.8 | 11.5 | | 57.1 | 11.6 | 0.79a | | 56.8 | 11.5 | | 57.1 | 11.6 | 0.79a |
| Histopathological subtypes | | | | | | | | 0.29b | | | | | | | 0.29b |
| Serous | 182 | 90 | | | 92 | | | | 90 | | | 92 | | | |
| Clear cell | 64 | 37 | | | 27 | | | | 37 | | | 27 | | | |
| Endometrioid | 43 | 17 | | | 26 | | | | 17 | | | 26 | | | |
| Mucinous | 14 | 6 | | | 8 | | | | 6 | | | 8 | | | |
| Other carcinoma | 17 | 10 | | | 7 | | | | 10 | | | 7 | | | |
| Non-epithelial cancer | 13 | | | | 13 | | | | | | | 13 | | | |
| Stage | | | | | | | | 0.97b | | | | | | | 0.97b |
| I | 82 | 39 | | | 43 | | | | 339 | | | 43 | | | |
| II | 33 | 15 | | | 18 | | | | 15 | | | 18 | | | |
| III–IV | 218 | 106 | | | 112 | | | | 104 | | | 112 | | | |
| Other cancers | 859 | 120 | | | 739 | | | | | | | | | | |
| BC | 115 | 15 | | | 100 | | | | | | | | | | |
| CC | 115 | 15 | | | 100 | | | | | | | | | | |
| ESCC | 88 | 15 | | | 73 | | | | | | | | | | |
| GC | 115 | 15 | | | 100 | | | | | | | | | | |
| HC | 81 | 15 | | | 66 | | | | | | | | | | |
| LC | 115 | 15 | | | 100 | | | | | | | | | | |
| PDA | 115 | 15 | | | 100 | | | | | | | | | | |
| SA | 115 | 15 | | | 100 | | | | | | | | | | |
| Borderline ovarian tumor | 66 | | | | 66 | | | | 33 | | | 33 | | | |
| Age, years | | | | | | 53 | 15.5 | | | 51.6 | 15.3 | | 54.3 | 15.9 | 0.50a |
| Histopathological subtypes | | | | | | | | | | | | | | | 0.85b |
| Serous | | | | | 18 | | | | 10 | | | 8 | | | |
| Mucinous | | | | | 32 | | | | 17 | | | 15 | | | |
| Granulosa cell | | | | | 5 | | | | 2 | | | 3 | | | |
| Others | | | | | 11 | | | | 4 | | | 7 | | | |
| Stage | | | | | | | | | | | | | | | 0.51b |
| I | | | | | 56 | | | | 26 | | | 30 | | | |
| II | | | | | 5 | | | | 3 | | | 2 | | | |
| III | | | | | 5 | | | | 4 | | | 1 | | | |
| Benign ovarian tumor | 29 | | | | 29 | | | | 15 | | | 14 | | | |
| Age, years | | | | | | 57.2 | 10.3 | | | 52.2 | 7.6 | | 62.6 | 10.3 | 0.004a |
| Histopathological subtypes | | | | | | | | | | | | | | | 0.73b |
| Serous | | | | | 5 | | | | 2 | | | 3 | | | |
| Mucinous | | | | | 11 | | | | 7 | | | 4 | | | |
| Others | | | | | 13 | | | | 6 | | | 7 | | | |
| Non-cancer controls | 115 | 15 | | | 100 | | | | 15 | | | 100 | | | |
| Age, years | | | 45.5 | 10.8 | | 45.6 | 9 | 0.97a | | 45.5 | 10.8 | | 45.6 | 9 | 0.97a |

BC breast carcinoma, CC colorectal adenocarcinoma, ESCC esophageal squamous cell carcinoma, GC gastric adenocarcinoma, HC hepatocellular carcinoma, LC lung carcinoma, PDA pancreatic ductal adenocarcinoma, SA bone and soft tissue sarcoma
aStudent's t-test
b$\chi^2$ test

associated with histopathological subtypes. Again, however, we could not assess this issue in the tissue miRNA data because all TCGA samples were of the serous adenocarcinoma subtype. Because the tissue miRNA clusters are associated with disease prognosis[25], further investigation of the association between circulating miRNA clusters and disease prognosis is warranted.

Our attempt to classify histopathological subtypes yielded little insight because these cancers did not have unique miRNA profiles, as determined by PCA mapping (Supplementary Figure 6). Because the tissue miRNA profile reflects the signature of the tumor origin[26], it may be generally true that the circulating miRNA profile cannot classify histopathological subtype in any cancer type. These ovarian cancer subtypes were reported to have genomic differences[27], but the DNAs or RNAs used for that study were mostly extracted from tumor tissue rather than bio-fluids. In addition, the percentage of circulating miRNAs in the bloodstream that are derived from tumors remains unknown, but it may have been quite low in the samples used in that research, potentially affecting the results. Even in this study, the sample size of each histopathological subtype was limited. Therefore, further large-scale validation is needed to determine whether histopathological subtypes specifically affect the circulating miRNA profile.

This study has some limitations. First, because it was performed using retrospectively collected samples, the processes before microarray analysis, such as the time interval between centrifugation and storage and the storage temperature, were not strictly regulated and may have differed between samples, as mentioned in the 'Methods' section. Although miRNAs are much more stable than mRNA, various processes can influence the levels of serum miRNAs[28,29]. Therefore, we collected non-cancer sample sets from three institutions and confirmed that diagnostic accuracy was maintained irrespective of the source. In addition, we have started a clinical prospective validation study, and we will be able to verify the generalizability of our data using fresh blood samples within a couple of years. Second, the absence of ovarian cancer was defined based

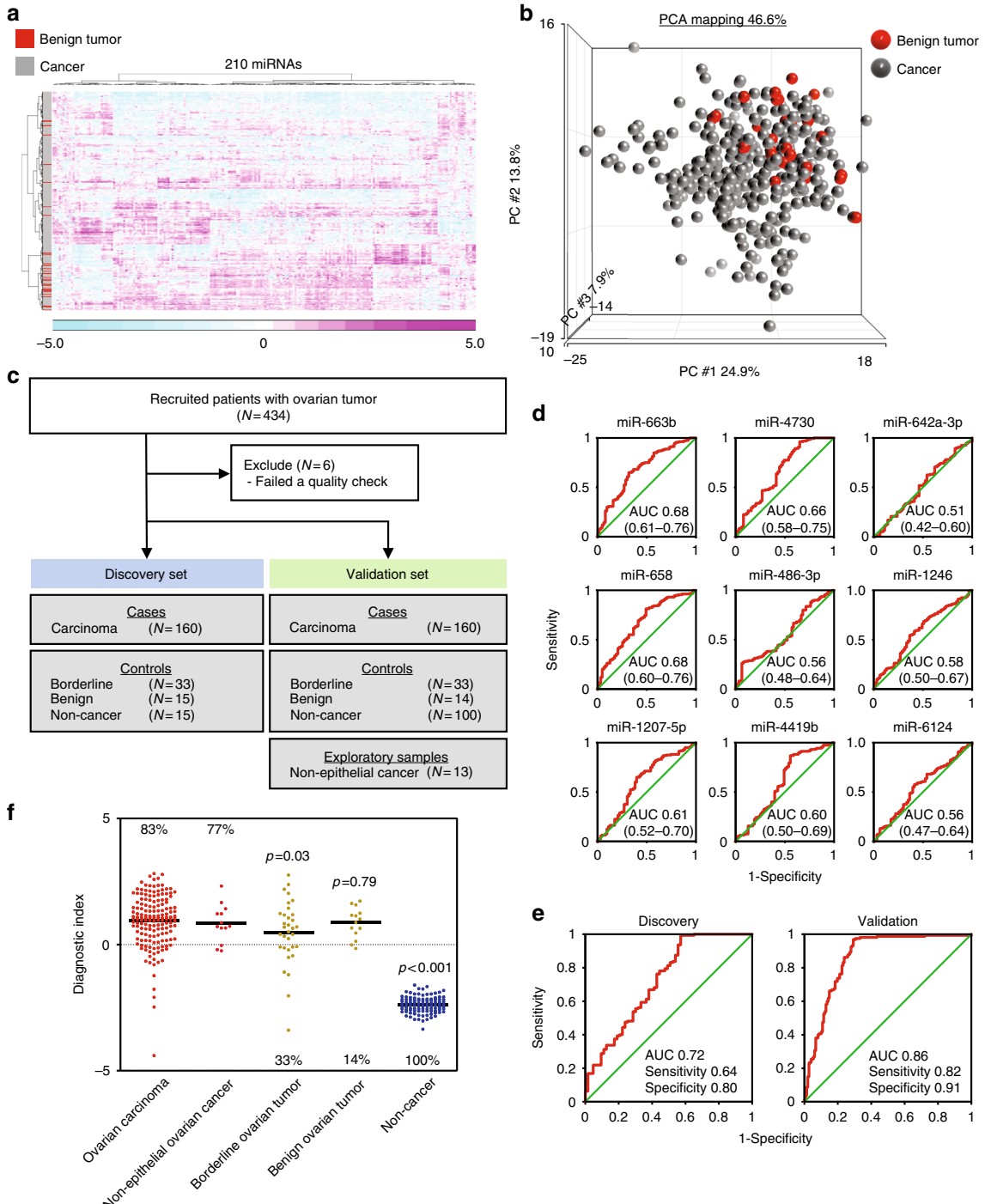

**Fig. 3** Development of the cancer-specific detection model (model 3). **a** Heatmap for serum miRNA expression of patients with benign tumors or cancer. $N$ = cancer, 333; benign tumors, 29. **b** PCA mapping for serum miRNA expression of patients with benign tumors or cancer. **c** Work flow of patients for development of prediction model 3. Serum samples were obtained from 543 subjects, including 320 patients with ovarian carcinoma, 66 with borderline tumors, 29 with benign tumors and 115 non-cancer controls (from non-cancer control B). The sample set was divided into two groups, the discovery set and validation set. **d** ROC curves for detecting cancer patients using miRNAs selected for prediction model 3. **e** Diagnostic performance of the selected nine miRNAs in the discovery set and validation set. **f** Cancer specificity of the diagnostic index using prediction model 3 in the validation set (ovarian cancer, 160; non-cancer [from non-cancer control B], 100; non-epithelial ovarian cancer, 13; borderline ovarian tumor, 33; and benign ovarian tumor, 14). Each diagnostic accuracy (%) is indicated. The $p$ values were calculated by $\chi^2$ test

on self-reported medical history in non-cancer control participants, but not precisely confirmed by gynecological examination.

In summary, our comprehensive analysis of serum miRNA profiles in 428 cases of ovarian tumors identified promising miRNA combinations for early-stage detection of ovarian cancer.

This kind of liquid biopsy represents a powerful tool in the clinical setting because the profile of a tumor is unstable and evolves dynamically over time, and tissue biopsies are almost impossible to obtain repeatedly[30]. On the other hand, we found that discrimination of ovarian cancer from borderline or benign

**Table 3 Independent association between the diagnostic indices and the presence of ovarian carcinoma**

|  | Odds ratio (95% CI)[a] | |
|---|---|---|
|  | Univariable analysis | Age-adjusted analysis |
| Model 1 | 12.3 (7.6–19.9) | 12.4 (7.7–20.1) |
| Model 2 | 9.6 (7.1–12.8) | 9.9 (6.8–12.1) |
| Model 3 | 3.6 (3.1–4.1) | 3.8 (3.2–4.4) |

*CI* confidence interval
[a]Logistic regression analysis

ovarian tumors using circulating miRNA profiles, or diagnosis of histopathological subtype, was more difficult than discrimination between cancer and non-cancer. Overall, our data suggest that evaluation of circulating miRNA is suitable for primary screening of ovarian cancer. The development of a less-invasive, rapid, and accurate diagnostic strategy for ovarian cancer even at an early stage should contribute to improvements in patient prognoses.

## Methods

**Clinical samples.** A total of 1496 serum samples were obtained from female patients with ovarian, breast, pancreatic, colorectal, hepatic, esophageal, gastric, and lung cancers, sarcoma, and non-cancer tumors (non-cancer control A) admitted or referred to the National Cancer Center Hospital (NCCH) between 2008 and 2016, and registered in the National Cancer Center (NCC) Biobank. Serum samples were stored at 4 °C for 1 week, and then stored at −20 °C until further use. Cancer patients with the following characteristics were excluded: (1) treatment with surgical operation, chemotherapy, or radiotherapy prior to serum collection, and (2) low-quality microarray data. Additional non-cancer control samples were obtained from the Yokohama Minoru Clinic (YMC) and the National Center for Geriatrics and Gerontology (NCGG). The first set (non-cancer control B) included 969 cancer-free female volunteers aged over 35 years who were recruited in 2015. The inclusion criteria for this sample set were no history of cancer and no hospitalization during the last 3 months, and the serum samples were collected and stored at −80 °C until further use. The second set (non-cancer control C) included 1581 individuals whose serum samples were collected between 2010 and 2012 and stored in the NCGG Biobank at −80 °C. Clinical information of all samples was obtained by referring to the registration information for each subject. Information about gynecological background, such as reproductive history or age at menopause, was not available for most samples. Due to insufficient descriptions, the histopathological subtypes of serous cancers contained a few low-grade tumors (<3%), but the vast majority of the population consisted of high-grade serous carcinoma (HGSC). The study was approved by the NCCH Institutional Review Board (2015-376, 2016-29) and the Research Ethics Committee of Medical Corporation Shintokai Yokohama Minoru Clinic (6019-18-3772). Written informed consent was obtained from each participant.

**miRNA expression arrays of clinical samples.** Total RNA was extracted from 300 μL of serum using the 3D-Gene® RNA extraction reagent (Toray Industries, Inc.). Comprehensive miRNA expression analysis was performed using the 3D-Gene® miRNA Labeling kit and the 3D-Gene® Human miRNA Oligo Chip (Toray Industries, Inc.), which was designed to detect 2588 miRNA sequences registered in miRBase release 21 (http://www.mirbase.org/). For quality control of microarray data, criteria for low-quality results were as follows: coefficient of variation for negative control probes >0.15; and number of flagged probes, identified by 3D-Gene® Scanner, >10; samples meeting these criteria were excluded from further analyses. The presence of miRNA was determined based on a corresponding microarray signal of greater than [mean + 2 × standard deviation] of the negative control signals, from which the most and least intense signals were removed. Once a miRNA was considered present, the mean signal of the negative controls (from which the top and bottom 5%, ranked by signal intensity, were removed) was subtracted from the miRNA signal. When the signal value was negative (or undetected) after background subtraction, the value was replaced by 0.1 on a base-2 logarithm scale. To normalize the signals among the microarrays tested, three pre-selected internal control miRNAs (miR-149-3p, miR-2861, and miR-4463) were used as previously described[31]. Each miRNA signal value was standardized using the ratio of the average signal value of the three internal control miRNAs to the pre-set value. All microarray data in the present study were obtained in accordance with the Minimum Information About a Microarray Experiment (MIAME) guidelines. TaqMan™ Advanced miRNA Assays (Thermo Fisher Scientific) were used for qRT-PCR validation. miRNA quantification data were normalized against the corresponding levels of miR-149-3p, miR-2861, and miR-4463. Serial dilutions of RNA extracted from human serum (LONZA, 14-490E) were used for quantitative performance assays (Supplementary Figure 2).The full miRNA expression

profiles are stored in the Gene Expression Omnibus (GEO) database (GSE106817, GSE103708).

**miRNA expression array in exosomes derived from cell lines.** Twelve human ovarian cancer cell lines were purchased from the American Type Culture Collection (ATCC; Manassas, VA, USA), the European Collection of Cell Cultures (ECACC; Porton Down, Wiltshire, UK), and the Japanese Collection of Research Bioresources (JCRB; Tokyo, Japan) cell bank (Supplementary Table 1) and all cells were certificated with no mycoplasma contamination. As previously described[16], all cell lines were cultured in optimal medium according to the suppliers' recommendations. The cells were washed with phosphate-buffered saline (PBS), and the culture medium was replaced with advanced Dulbecco's modified Eagle's medium for ES-2, SKOV3, CAOV3, OV-90, OAW42, COV362, and MCAS cells, advanced DMEM/Ham's F-12 medium for RMG-1 and RUG-S cells, or advanced RPMI medium for A2780, OVCAR3 and KURAMOCHI cells. After incubation for 48 h, the conditioned medium (CM) was collected and centrifuged at 2000 × g for 10 min at 4 °C. To thoroughly remove cellular debris, the supernatant was filtered through a 0.22-μm filter (Millipore). To prepare exosomes, CM was ultracentrifuged at 35,000 rpm using a SW41Ti rotor for 70 min at 4 °C. The pellets were washed with PBS, ultracentrifuged at 35,000 rpm using the SW41Ti rotor for 70 min at 4 °C and resuspended in PBS. Total RNA was extracted from those exosomes using QIAzol and the miRNeasy Mini Kit (Qiagen, Hilden, Germany) as instructed by the manufacturer's protocols. Total RNA was labeled with cyanine 3 (Cy3) using the miRNA Complete Labeling and Hyb Kit (Agilent Technologies) as instructed by the manufacturer. Agilent SurePrint G3 Human miRNA 8 × 60 K Rel.19 (design ID: 046064) arrays were used and scanned using an Agilent DNA microarray scanner. The intensity values for each scanned feature were quantified using Agilent Feature Extraction software version 10.7.3.1. The expression analysis was performed with Agilent GeneSpring GX version 13.0.

**Statistical analysis.** Prior to statistical comparisons, samples were divided into discovery and validation sets. The discovery set was used to select miRNA markers and construct discriminant models, and the validation set was used to validate the discriminant models.

Based on combinatorial optimization for multicandidate miRNAs, diagnostic indices were generated using Fisher's linear discriminant analysis. Using leave-one-out cross-validation in the discovery set, the best diagnostic index was ultimately selected. An index score ≥ 0 indicated ovarian cancer, and an index score <0 indicated the absence of ovarian cancer, including the presence of other cancers and non-cancer controls. The diagnostic sensitivity, specificity, accuracy, and area under the receiver operating characteristics (ROC) curve (AUC) were calculated for each diagnostic index in the validation sets.

Statistical analyses were performed using R version 3.1.2 (R Foundation for Statistical Computing, http://www.R-project.org), compute.es package version 0.2–4, glmnet package version 2.0–3, hash package version 2.2.6, MASS package version 7.3–45, mutoss package version 0.1–10, pROC package version 1.8, and IBM SPSS Statistics version 22 (IBM Japan, Tokyo, Japan). Unsupervised clustering and heatmap generation with sorted datasets, using Pearson's correlation in Ward's method for linkage analysis, were performed using Partek Genomics Suite 6.6. PCA was also performed using Partek Genomics Suite 6.6. The limit of statistical significance for all analyses was defined as a two-sided p value of 0.05.

## Data availability

The microarray data that support this study are available through the NCBI database under accession GSE106817 and GSE103708. All other relevant data are available within the article file or Supplementary Information, or available from the authors on reasonable request.

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

## Acknowledgements

The authors thank Tomomi Fukuda, Hiroko Tadokoro, Tatsuya Suzuki, Makiko Ichikawa, Junpei Kawauchi, Satoshi Kondou, and Kamakura Techno-Science Inc. for performing the microarray assays. The authors thank Noriko Abe and Michiko Ohori for collecting samples from the freezing room and Kazuki Sudo for independent confirmation of participant eligibility. Some of the samples and clinical information used in this study was obtained from the National Cancer Center Biobank, which is supported by National Cancer Center Research and Development Fund (29-A-1). The authors also thank the Biobank at the National Center for Geriatrics and Gerontology for providing biological resources. This study was financially supported through a Development of Diagnostic Technology for Detection of miRNA in Body Fluids grant from the Japan Agency for Medical Research and Development (to TO).

## Author contributions

A.Y., J.M. and T.O. designed the experimental approach. A.Y., J.M., Y.Ya., T.S., J.K., S.T. and Y.A. performed the experiments and analyzed the data. A.Y., J.M. and T.O. wrote the manuscript, and Y. Ya assisted. Y.Yo., K.T., H.S., T.U., M.I., S.I., S.N., H.S., K.K. and T.K. managed and provided the patient samples. The manuscript was finalized by T.O. with the assistance of all the authors.

## Additional information

**Competing interests:** J.K. and S.T. are employees of Toray Industries, Inc., the provider of the 3D-Gene® system. Y.A. is an employee of Dynacom Co., Ltd., the developer of the statistical script used to select the best miRNA combination. The remaining authors declare no competing interests.

