## [Peer Review File · Nature Communications]

Reviewers' comments:

Reviewer #1 (Remarks to the Author):

In this study, the authors present a retrospective analysis of miRNA isolated from serum of 434 women with ovarian cancer (of all subtypes and stages). They present three models for use of serum miRNA as a potential tool to screen women for the presence of ovarian cancer. The comparators include serum collected from nearly 2800 healthy volunteers as well as serum samples from a small number of patients with non-malignant ovarian disease (non-epithelial ovarian cancers, benign and borderline ovarian tumours) as well as from women with other solid tumours. The analysis platform is array-based, assessing 2588 miRNA, but with no validation using either RT-PCR or RNASeq.

Three separate models are presented.

Model 1 – 210 miRNA were identified using analysis of exosomes released from 12 ovarian cancer cell lines. Using discovery cohort of 160 ovarian cancer patients (all stages and subtypes) plus 1379 normal controls, this was further reduced to 6; miR-5100, miR-4532, miR-614, miR-1233-5p, miR-4697-5p, and miR-451a, which were then assessed in a validation cohort of another 160 ovarian cancer patients, with 13 non-epithelial, 66 borderline and 29 benign. This model revealed excellent (in fact, perfect) discrimination between cancer and benign, but could not differentiate non-epithelial or borderline or benign ovarian tumour. There was no difference in the cancer cases across stage or subtype of invasive disease.

In model 2, the authors looked at miRNA in serum from women with a wide variety of solid malignancies, including breast cancer, PDAC, colorectal, HCC, bone and soft tissue sarcoma. 15 of each of these non-ovarian cancer cases were utilised in the discovery set, with the remaining (66 – 100 cases) into the validation set. There were also non-epithelial ovarian cancers plus borderline and benign ovarian tumours in the validation set as well. Nine miRNA emerged from this analysis: miR-4739, miR-939-5p, miR-1469, miR-1273g-3p, miR-4697-5p, miR-3197, miR-6717-5p, miR-4726-5p, and miR-23a-3p, ie only one of which was in common with model 1 (miR-4697-5p). Performance in discovery was AUC 0.95 and 0.88 in validation set. However, model 2 was still unable to differentiate non-epithelial ovarian tumours, borderline and benign ovarian tumours, plus half of sarcomas.

In model 3, PCA and cluster analysis was used to differentiate between cancer and benign tumours. This highlighted another set of miRNA: hsa-miR-1273g-3p, hsa-miR-135a-3p, hsa-miR-4443, hsa-miR-1225-5p, hsa-miR-3135b, hsa-miR-4516, hsa-miR-1225-3p, hsa-miR-6724-5p, and hsa-miR-4476, ie no crossover with the two previous datasets. AUC in discovery set was 0.79 and 0.89 in validation set. Model 3 could still not distinguish between ovarian cancer and non-epithelial tumour, borderline and benign ovarian cancers, although was very good indeed at discriminating non-cancer. Interestingly, the non-epithelial ovarian cancers and the non-invasive tumours did not have distinct expression profiles.

Across all three models, the non-cancer control group were younger than the cancer cohorts, but adjusted for age, all three models could predict presence of ovarian cancer adjusted for age. The sample collection SOP appears to have been different for these non-cancer control samples, which were collected over a much shorter time period.

Interestingly, when the ovarian cancer subtypes were analysed separately, PCA mapping showed no differences, whilst unsupervised clustering of miRNA in ovarian cancer cases identified three clusters that spanned across all subtypes. Interestingly, TCGA analysis of tumour-associated miRNA also identified three clusters – however, there are few details publicly available to allow cross-comparison with the three clusters identified here. In addition, TCGA was limited to high grade serous carcinoma.

Overall, this is interesting work that addresses an important clinical problem. There are several important questions, however, that remain:

1. What is the overall purpose of the miRNA analysis? Is it to test asymptomatic women as a population-based screen? Or is it to improve diagnostic accuracy (cancer vs benign) in those already referred to a gynaecological cancer centre? This is a very important point because the performance characteristics required differ significantly in the two situations. Reading this manuscript, I am unclear where the authors believe miRNA analysis fits.
2. Methods 1 - miRBASE version 21 contains over 28450 miRNA, so how were the 2588 miRNA on the array selected? Also, I disagree with the authors that it is not important to validate specific miRNA using separate technology eg RT-PCR or RNAseq (lines 300 – 301) – given that only a small number of miRNA were used in the three models, I believe that it is both simple and important to validate.
3. Methods 2. I have some concerns that serum samples from cancer patients were stored at 4 degrees for one week prior to transfer to -20. Most protocols would require clotting for a short time, followed by storage at -80 – indeed, the non-cancer controls were stored at -80. Thus, there is no consistent SOP for collection, processing and storage. Do the authors believe that this makes a significant difference?
4. Methods 3. The authors will be aware that there is concern over the suitability of some cell lines as models of high grade serous ovarian cancer (Domcke et al Nature Communications 2013). In particular, SKOV3 and A2780 were thought to be unlikely to be of HGSC subtype. Does this study not need to be performed only using HGSC cell lines (e.g. primary cultures and/or known HGSC lines) and HGSC tumours?
5. Ovarian cancer biology. The authors state that tissue miRNA profile reflects tissue of origin and hence why it isn't possible to distinguish a) between invasive and borderline and b) between ovarian cancer subtypes. However, this slightly betrays misunderstanding of ovarian cancer biology – indeed, tables 1 and 2 list only 'serous' carcinoma: for the past decade, serous carcinoma has been divided into high grade and low grade, which are essentially two separate diseases. Grouping them together does not make biological sense. I am intrigued that the different histological subtypes do not generate separate miRNA, given that they are such very different diseases – high grade serous carcinoma probably arises in the distal fallopian tube and has more in common with triple negative breast cancer than it does with low grade endometrial carcinoma or clear cell carcinoma of the ovary.

Reviewer #2 (Remarks to the Author):

The manuscript describes about the use of circulating miRNA as biomarkers for diagnosis of ovarian cancer patients. The study is designed well with decent number of samples that is required for biomarker analysis. However, the paper could have been simplified more especially in the patients and miRNAs selected. From my limited experience in such studies, the data presented and the experiments performed seems to be of high quality. The authors need to address the following points to improve the manuscript.

1. As from the current manuscript, the whole study is based on microarray expression profile of serum samples. With the identified models (combinatorial biomarker candidates), it would be great if the authors perform a bling test validation by qPCR or digital PCR analysis among the serum samples. These results are important for the clinical utility of the study.
2. Page 6, section on selection of circulating miRNA biomarker candidates is not clear. Even after repeated reading, it is hard to understand what the authors have done. Detailed explanation is required to do justice for this section.
3. For Supplementary Figure 1, were the 858 miRNAs detected at least in 1 EV sample? What about the 210 miRNAs? Were they detected in at least 1 patient sample or more than 10 patient samples? What is the cut-off here?
4. The patient numbers in Figure 1 and Table 1 does not add up. Doesn't Validation set contain

173 patients as compared to 160 in Figure 1? Also, how is 428 patients distributed in discovery and validation sets?

5. How was the nine miRNAs selected for model 2? Was these miRNAs in EVs?

Reviewer #3 (Remarks to the Author):

The authors tackle a highly important yet challenging clinical scenario: to be able to detect ovarian cancers before they expand to stage 3 or 4 disease where cancer control becomes more challenging. They seek to achieve this through another ambitious effort: the liquid biopsy. A strength here is the precious biorepository with numerous cancer patients across subtypes. The effective use of such specimens along with robust clinical documentation would be poised to expand our scientific understanding of ovarian cancer. However, the weaknesses noted here appear to outweigh the strengths.

There was a lack of clarity about a priori hypotheses vis-a-vis selected biomarkers in relation to ovarian cancers. The parent project seems to be an ambitious large scale effort to analyze miRNA profiles across various cancer types. The message transmitted was one of data fishing and forced application of modeling to arrive at distinct signatures to discriminate cancers from other subtypes or benign. In the end, even the data is not clinically actionable and remains to be prospectively validated - which to the authors credit is currently under way. For example, Model 1 could not distinguish ovarian cancer from benign masses; a key deficit in a screening population. Model 2 could not distinguish ovarian cancer from esophageal cancers or sarcomas - two entities with drastically different management approaches compared to ovarian cancer. Model 2 also fared poorly with benign masses. A third attempt (Model 3) showed "marginal" improvement but not against borderline tumors. The sensitivity here was 64% and hence more like negligible rather than marginal. miRNA analyses did not differ among all four ovarian cancer subtypes; this is difficult to believe given the known genomic differences between clear cell ovarian cancer and their other histology counterparts.

The study also appeared to take a brief detour towards exosomes that appears forced as well. Here, only cell lines are used and not clinical specimens. It would be difficult to analyze serum using human samples given that most of the field is gravitating towards plasma. No innovation here was noted since (based on the reference they cite in their methods) ultracentrifugation [conventional approach] and commercially available kits were used to analyze exosomes. There is an old adage: garbage in, garbage out that resonates with those who work in the biomarker space. Notably, concerns about pre-analytical handling can create downstream unreliability. The authors also alluded to this lack of uniform handling which can impact reproducibility by others and readouts. For instance, perhaps some of the readouts they noted were merely artifacts of specimen handling and/or storage (e.g. repeat freeze-thaw).

For now, this is another biomarker study that needs validation; no new technology or innovation is presented here. Pairing the work with one of the various forward thinking technologies out there (including those in the exosome space) could create more traction. No info presented here would move the clinical needle in the near or intermediate term. I would be excited, however, to see the results of their prospective trial which I assume would also contain highly standardized pre-analytical standard operating procedures. A more biomarker focused journal may be more suitable in the current state.

Reviewer #1

We appreciate your critical comments, which helped us to improve the manuscript. All of the points you raised have been addressed in the revision.

Comment_1: *What is the overall purpose of the miRNA analysis? Is it to test asymptomatic women as a population-based screen? Or is it to improve diagnostic accuracy (cancer vs benign) in those already referred to a gynaecological cancer centre? This is a very important point because the performance characteristics required differ significantly in the two situations. Reading this manuscript, I am unclear where the authors believe miRNA analysis fits.*

Response: The main purpose of this study was to develop the best screening method for ovarian cancer patients; currently, an optimized method for ovarian cancer screening does not exist. However, we also wanted to make the most of data from miRNA microarray analyses. For that reason, we tried to analyze and develop additional predictive models that are clinically valuable, as you mentioned in the second question about the purpose of the work. Unfortunately, the results of the additional models were not perfectly concordant with those of the first model. To address your concerns, we added the following sentences to clarify the purpose of the study.

Page 5

“Our primary aim is to develop a novel screening strategy capable of discriminating cancer patients from healthy women. In addition, comprehensive profiles of circulating miRNAs, which were obtained from all samples, enabled us to generate an optimal diagnostic model for ovarian cancer.”

Comment_2: *Methods 1 - miRBASE version 21 contains over 28450 miRNA, so how were the 2588 miRNA on the array selected? Also, I disagree with the authors that it is not important to validate specific miRNA using separate technology eg RT-PCR or RNAseq (lines 300 = 301) = given that only a small number of miRNA were used in the three models, I believe that it is both simple and important to validate.*

Response: The number of entries in miRBASE (28,450) includes miRNAs from a broad range of species. A total of 2588 human miRNAs are currently recognized. We appreciate your suggestion, however, and revised part of the discussion accordingly. To address your concerns, we performed qPCR validation in a small sample set, and modified the text accordingly, as follows:

“First, we identified the best 10 miRNAs with the highest AUC values, termed “pivot miRNAs,” as shown in Supplementary Table 2. A predictive model was created based on these pivot miRNAs, and other miRNAs were used to compensate for their diagnostic performance. Thus, the levels of the pivot miRNAs are the most important factors governing model performance, and must be reproducible on independent platforms if these models are to proceed to clinical application. Accordingly, we investigated the expression of pivot miRNAs by qRT-PCR. Around 60% of miRNAs were detectable in the quantitative performance assay and were used for subsequent analysis (Supplementary Figure 2); an R-value of 0.9 was considered as cut-off value for inclusion. Then, to assess reproducibility of miRNAs between microarray and qRT-PCR analyses, the levels of each miRNA were plotted as histograms using ten patient samples randomly selected from the non-cancer A (N = 5) and ovarian cancer cohorts (N = 5), as shown in Supplementary Figure 3. If the R-values were less than zero, we considered the miRNA to be validated by qRT-PCR.”

We then adjusted all models based on highly reproducible pivot miRNAs. We believe that these improvements allow the models to be upgraded and adapted to other applications.

Supplementary Figure 2

Supplementary Figure 3

Comment_3: *Methods 2.* I have some concerns that serum samples from cancer patients were stored at 4 degrees for one week prior to transfer to -20. Most protocols would require clotting for a short time, followed by storage at -80 — indeed, the non-cancer controls were

stored at -80. Thus, there is no consistent SOP for collection, processing and storage. Do the authors believe that this makes a significant difference?

Response: As you mentioned, the storage procedures were not consistent among facilities, and we consider this a limitation of this study. On the other hand, we observed a strict SOP from sample thawing to data generation, which we consider a strong point of the study. Unfortunately, we do not have sufficient evidence about whether the difference in storage conditions affect the results, but in general, miRNAs are stable in bio-fluids due to their small size and protection by packaging in exosomes^{1,2}. In addition, we prepared non-cancer A cohort, which is the same condition of cancer cohort. Even though the non-cancer A set consisted of benign tumors, they could be successfully distinguished from cancers. For these reasons, we concluded that differences between profiles were not due to storage conditions, but rather to the innate features of the samples. Sample storage is now addressed in the Discussion section.

“Although miRNAs are much more stable than mRNA, various processes can influence the levels of serum miRNAs. Therefore, we collected non-cancer sample sets from three institutions and confirmed that diagnostic accuracy was maintained irrespective of the source. In addition, we have started a clinical prospective validation study, and we will be able to verify the generalizability of our data using fresh blood samples within a couple of years.”

Comment_4: *Methods 3. The authors will be aware that there is concern over the suitability of some cell lines as models of high grade serous ovarian cancer (Domcke et al Nature Communications 2013). In particular, SKOV3 and A2780 were thought to be unlikely to be of HGSC subtype. Does this study not need to be performed only using HGSC cell lines (e.g. primary cultures and/or known HGSC lines) and HGSC tumours?*

Response: We are aware of these concerns. Before preparing 13 cell lines, we carefully referred to the cited report (Domcke et al., *Nature Communications* 2013) and selected cells favorable for HGSC models, such as KURAMOCHI and COV362, for inclusion in our lists.

However, we did not want to choose only HGSCs, but also other subtypes, because the patient sample set also included various subtypes, and the main purpose of this study was to develop screening methods. No clinician can determine the type of an ovarian tumor before biopsy. For this reason, we needed to select a broad range of candidate miRNAs that could be related to ovarian cancer.

Comment_5: *Ovarian cancer biology. The authors state that tissue miRNA profile reflects tissue of origin and hence why it isn't possible to distinguish a) between invasive and borderline and b) between ovarian cancer subtypes. However, this slightly betrays misunderstanding of ovarian cancer biology = indeed, tables 1 and 2 list only 'serous' carcinoma: for the past decade, serous carcinoma has been divided into high grade and low grade, which are essentially two separate diseases. Grouping them together does not make biological sense. I am intrigued that the different histological subtypes do not generate separate miRNA, given that they are such very different diseases = high grade serous carcinoma probably arises in the distal fallopian tube and has more in common with triple negative breast cancer than it does with low grade endometrial carcinoma or clear cell carcinoma of the ovary.*

Response: We were slightly disappointed by the observation that circulating miRNA profiles in serum did not reflect ovarian cancer background. However, although this is a negative result, we felt it was important to provide readers with this information. As you mentioned, genomic differences among other ovarian cancer subtypes have been reported, but the DNAs or RNAs used for those studies were mostly extracted from tumor tissue rather than bio-fluids. In addition, the percentage of circulating miRNAs in the bloodstream that are derived from tumors remains unknown, but it may have been quite low in the samples used in that research, potentially affecting the results. These points were added to the Discussion section, as follows:

"Distinguishing benign tumors from malignant cancers is a major concern for gynecologists, and a less-invasive diagnostic method would be of great clinical value. Our model 3 could not discriminate ovarian cancer from benign tumors (Figure 3f). Several previous reports

using much smaller sample sets suggested that it would be possible to discriminate malignant from benign ovarian tumors using circulating miRNAs³⁻⁵. However, none of those studies reported diagnostic performance in terms of sensitivity and specificity, and no previous study has developed a model capable of discriminating between malignant and borderline ovarian tumors. In this study, we also found it difficult to discriminate between ovarian cancer and borderline ovarian tumors. However, because the sample sizes of borderline or benign ovarian tumors were insufficient for model construction, further large-scale validation studies are needed to resolve this issue.”

We understand that LGSC and HGSC are completely different diseases. In this study, the number of LGSC patients was less than 3% in the “serous” group, and some cases, especially in those from the past, were not accompanied by a description of tumor grades. For that reason, it was hard for us to correctly separate LGSC from HGSC. To avoid misunderstandings, the following sentences were added to the revised manuscript:

“Due to insufficient descriptions, the histopathological subtypes of serous cancers contained a few low-grade tumors (<3%), but the vast majority of the population consisted of high-grade serous carcinoma (HGSC).”

To address the biology of differences among subtypes, it is necessary to analyze primary tumors as well, and to prepare the best cell lines and mouse models for validation experiments. We very much appreciate these comments, and we plan on investigating these issues in future studies.

Reviewer #2

We appreciate your critical comments, which have helped us improve the manuscript. All of the points that you raised have been addressed in the revision.

Comment_1: *As from the current manuscript, the whole study is based on microarray expression profile of serum samples. With the identified models (combinatorial biomarker candidates), it would be great if the authors perform a bling test validation by qPCR or digital PCR analysis among the serum samples. These results are important for the clinical utility of the study.*

Response: To confirm the reproducibility of the selected miRNAs on other platforms, we performed qPCR validation in small sample set, as described in the following text:

“First, we identified the best 10 miRNAs with the highest AUC values, termed “pivot miRNAs,” as shown in Supplementary Table 2. A predictive model was created based on these pivot miRNAs, and other miRNAs were used to compensate for their diagnostic performance. Thus, the levels of the pivot miRNAs are the most important factors governing model performance, and must be reproducible on independent platforms if these models are to proceed to clinical application. Accordingly, we investigated the expression of pivot miRNAs by qRT-PCR. Around 60% of miRNAs were detectable in the quantitative performance assay and were used for subsequent analysis (Supplementary Figure 2); an R-value of 0.9 was considered as cut-off value for inclusion. Then, to assess reproducibility of miRNAs between microarray and qRT-PCR analyses, the levels of each miRNA were plotted as histograms using ten patient samples randomly selected from the non-cancer A (N = 5) and ovarian cancer cohorts (N = 5), as shown in Supplementary Figure 3. If the R-values were less than zero, we considered the miRNA to be validated by qRT-PCR.”

We then adjusted all models based on highly reproducible pivot miRNAs. We believe that these improvements allow the models to be upgraded and adapted to other applications.

Supplementary Figure 2

Supplementary Figure 3

Comment 2: Page 6, section on selection of circulating miRNA biomarker candidates is not clear. Even after repeated reading, it is hard to understand what the authors have done. Detailed explanation is required to do justice for this section. For Supplementary Figure 1, were the 858 miRNAs detected at least in 1 EV sample? What about the 210 miRNAs? Were

they detected in at least 1 patient sample or more than 10 patient samples? What is the cut-off here?

Response: We apologize for the overly complicated description. Actually, the 858 miRNAs were detected in exosomes from at least in one ovarian cancer cell line, indicating that they could be related to ovarian cancer. No cut-off was used in this case; instead, we followed the “all-or-non” principle. Among these 858 miRNAs, 648 were not detected in serum from 4046 patients. As described in the Methods section, highly expressed miRNAs in patient samples was determined based on a corresponding microarray signal of $>2^6$ in more than 50% of samples. Ultimately, a final set of 210 miRNAs had sufficient signals in patient samples, and could be confidently classified as miRNAs secreted from ovarian cancer cells. We considered these candidates to be the most promising. To clarify, we rewrote this part of the text and revised Supplementary Figure 1.

“To focus on extracellular miRNAs released from ovarian cancer cells, we evaluated miRNA expression in EVs, including exosomes, from 12 ovarian cancer cell lines (listed in Supplementary Table 1). A total of 858 miRNAs detected in exosomes derived from at least one cell line were considered as candidate ovarian cancer-released miRNAs. We proceeded to further select miRNA candidates using a human serum dataset. Specifically, miRNAs with a signal value $>2^6$ in more than 50% of samples were selected as robust biomarkers in serum samples from ovarian cancers. Based on this analysis, 648 miRNAs (of the previously selected 858) were excluded due to low signal. Ultimately, 210 miRNAs were selected for further analyses (Supplementary Figure 1).”

Supplementary Figure 1

Comment_3: *The patient numbers in Figure 1 and Table 1 does not add up. Doesn't Validation set contain 173 patients as compared to 160 in Figure 1? Also, how is 428 patients distributed in discovery and validation sets?*

Response: We appreciate your pointing this out. We carefully checked this part of the manuscript and revised Figure 1a and Table 1. Actually, non-epithelial ovarian cancer cells were included in the Validation set in Table 1. As stated on page 6, the patients were randomly assigned to two groups.

Comment_4: *How was the nine miRNAs selected for model 2? Was these miRNAs in EVs?*

Response: The selection methods for all three models were the same (Fisher's linear discriminant analysis), as described on pp. 7–8. We designed comprehensive discriminants consisting of one to ten miRNAs from the discovery set. Based on the optimal level of accuracy (Supplementary Table 3, 5, and 7), this analysis provided the best discrimination within the discovery set. As explained in the response to Comment 2, all miRNAs in this analysis were package in EVs derived from at least one ovarian cancer cell line, but we did not confirm their presence in EVs derived from patients serum. We very much appreciate these comments, and hope to investigate these issues in future studies.

Reviewer #3

We really appreciate your many critical and valuable comments, which helped us to improve our study. All of the points that you raised have been addressed as described below.

Comment_1: *There was a lack of clarity about a priori hypotheses vis-a-vis selected biomarkers in relation to ovarian cancers. The parent project seems to be an ambitious large scale effort to analyze miRNA profiles across various cancer types. The message transmitted was one of data fishing and forced application of modeling to arrive at distinct signatures to discriminate cancers from other subtypes or benign. In the end, even the data is not clinically actionable and remains to be prospectively validated - which to the authors credit is currently under way.*

Response: As stated in the Introduction part, the lack of an effective screening method for ovarian cancer was the primary motivation for designing this study. Initially, we set out to test the hypothesis that circulating miRNAs can serve as screening biomarkers. Until now, their potential in this regard has remained unclear because no large-scale analyses have focused on extracellular nucleic acids. Our main conclusion is that miRNAs indeed represent promising biomarkers for screening method capable of distinguishing cancer patients from healthy women. Further prospective validation is definitely necessary, and already underway, but we believe that the results of this study can shed light on problems related to ovarian cancer screening. The pertinent section of the Introduction was revised as follows:

“Our primary aim is to develop a novel screening strategy capable of discriminating cancer patients from healthy women. In addition, comprehensive profiles of circulating miRNAs, which were obtained from all samples, enabled us to generate optimal diagnostic models for ovarian cancer.”

Comment_2: *For example, Model 1 could not distinguish ovarian cancer from benign masses; a key deficit in a screening population. Model 2 could not distinguish ovarian cancer from esophageal cancers or sarcomas - two entities with drastically different management*

approaches compared to ovarian cancer. Model 2 also fared poorly with benign masses. A third attempt (Model 3) showed "marginal" improvement but not against borderline tumors. The sensitivity here was 64% and hence more like negligible rather than marginal. miRNA analyses did not differ among all four ovarian cancer subtypes; this is difficult to believe given the known genomic differences between clear cell ovarian cancer and their other histology counterparts.

Response: The concerns pointed out in this comment are indeed unfavorable or negative results of this study, but we felt that it was important to provide this information to the reader. As you mentioned, it was difficult to accurately distinguish ovarian cancer from benign masses. However, the purpose of model 1 is to test asymptomatic women in the context of a population-based screen. For this reason, it is not disadvantageous to judge women with benign masses as positive, because such patients can be encouraged to undergo clinician follow-up; indeed, in some cases, some benign masses should be treated surgically. Diagnoses of cancer should be performed by multimodal assessment, e.g., examination by clinicians in conjunction with imaging by ultrasound, CT, or MRI. The worst-case scenario of this model would be a high rate of false-negative case, but this was not observed. These points are now addressed in the Discussion section, as follows:

"In this study, we also found it difficult to discriminate between ovarian cancer and borderline ovarian tumors. However, because the sample sizes of borderline or benign ovarian tumors were insufficient for model construction, further large-scale validation studies are needed to resolve this issue."

It is hard to say why model 2 could not distinguish ovarian cancer from esophageal cancers or sarcomas. Currently, we cannot explain this result. However, the similarities among those types of cancers should be investigated in future studies.

As you mentioned, it was also difficult to identify patients with borderline tumors. We deleted the term "marginal" from the manuscript, and addressed the reasons in the Discussion. Specifically, we attributed this feature to the lack of unique expression profiles, as shown in Supplementary Figure 2, and the size of sample cohort. It remains challenging

to distinguish these tumor types.

Regarding the lack of a difference among all four ovarian cancer subtypes (page 11), as you mentioned, genomic differences among other ovarian cancer subtypes have been reported, but the DNAs or RNAs used for those studies were mostly extracted from tumor tissue rather than bio-fluids. In addition, the percentage of circulating miRNAs in the bloodstream that are derived from tumors remains unknown, but it may have been quite low in the samples used in that research, potentially affecting the results. These points were added to the Discussion section, as follows:

“These ovarian cancer subtypes were reported to have genomic differences ⁶, but the DNAs or RNAs used for that study were mostly extracted from tumor tissue rather than bio-fluids. In addition, the percentage of circulating miRNAs in the bloodstream that are derived from tumors remains unknown, but it may have been quite low in the samples used in that research, potentially affecting the results.”

Comment_4: *The study also appeared to take a brief detour towards exosomes that appears forced as well. Here, only cell lines are used and not clinical specimens. It would be difficult to analyze serum using human samples given that most of the field is gravitating towards plasma. No innovation here was noted since (based on the reference they cite in their methods) ultracentrifugation [conventional approach] and commercially available kits were used to analyze exosomes.*

Response: In this study, we used information from miRNA profiles in exosomes to confirm whether candidate miRNAs were secreted from ovarian cancer cells. If the miRNAs in our models were not released from such cells, they could distract focus from relevant findings. For this reason, exosomes should be isolated by the method recommended by the International Society of Extracellular vesicles ⁷, and we did not try to innovate any novel methods. As you mentioned, it is important to analyze exosomal miRNAs in human serum or plasma, but in this study, we had to work within limitations on sample volume, experimental procedures, and storage condition. All samples in our biobank were stored as serum, and the

current method for isolation of exosomes is not sufficiently high-throughput. In addition, higher volumes of samples are generally required for microarray analysis. These concerns were addressed in the Discussion section as follows:

“In the current study, we did not isolate exosomes from serum due to limitations on sample volume; moreover, such isolation is not compatible with high throughput. However, it is possible that miRNAs are packaged in exosomes and that they might play functional roles.”

Comment_5: *There is an old adage: garbage in, garbage out that resonates with those who work in the biomarker space. Notably, concerns about pre-analytical handling can create downstream unreliability. The authors also alluded to this lack of uniform handling which can impact reproducibility by others and readouts. For instance, perhaps some of the readouts they noted were merely artifacts of specimen handling and/or storage (e.g. repeat freeze-thaw). For now, this is another biomarker study that needs validation; no new technology or innovation is presented here. Pairing the work with one of the various forward thinking technologies out there (including those in the exosome space) could create more traction. No info presented here would move the clinical needle in the near or intermediate term. I would be excited, however, to see the results of their prospective trial which I assume would also contain highly standardized pre-analytical standard operating procedures. A more biomarker focused journal may be more suitable in the current state.*

Response: As you mentioned, the storage procedures were not consistent among facilities, and we consider this a limitation of this study. On the other hand, we observed a strict SOP from sample thawing to data generation, which we consider a strong point of the study. Unfortunately, we do not have sufficient evidence about whether the difference in storage conditions affect the results, but in general, miRNAs are stable in bio-fluids due to their small size and protection by packaging in exosomes^{1,2}. In addition, we prepared non-cancer A cohort, which is the same condition of cancer cohort. Even though the non-cancer A set consisted of benign tumors, they could be successfully distinguished from cancers. For these reasons, we concluded that differences between profiles were not due to storage conditions,

but rather to the innate features of the samples. Accordingly, we are confident that we did not analyze garbage. Sample storage is now addressed in the Discussion section:

“First, because it was performed using retrospectively collected samples, the processes before microarray analysis, such as the time interval between centrifugation and storage and the storage temperature, were not strictly regulated and may have differed between samples, as mentioned in the Methods section. Although miRNAs are much more stable than mRNA, various processes can influence the levels of serum miRNAs. Therefore, we collected non-cancer sample sets from three institutions and confirmed that diagnostic accuracy was maintained irrespective of the source. In addition, we have started a clinical prospective validation study, and we will be able to verify the generalizability of our data using fresh blood samples within a couple of years.”

In addition, to address the reviewer’s concern about reproducibility, we performed qRT-PCR experiments to validate the results and identify highly reproducible miRNAs (Supplementary Figure 2–3) on independent platforms, and then adjusted all three models. Because the models can reproduce results obtained using other platforms, we believe that these improvements will allow the models to be upgraded and adapted for additional applications. Further prospective validation is definitely necessary, and is already underway, with highly standardized pre-analytical SOP, but we think that this should be published as an independent study. We believe that the existing data are sufficient to merit publication now.

We grant that the study contained no new technology or innovation, but we believe that the statistical approach to identifying candidate miRNAs represents a kind of new methodology, and that the insights we provide regarding the interpretation of extracellular miRNA profiles at such large scale should also be considered novel information. Related functional analysis might spread by focusing on pivot miRNAs, as shown in Supplementary Tables 2, 4, and 6. In this revised version, we also confirmed the reproducibility of those miRNAs by qRT-PCR (Supplementary figure 3). Some miRNAs have already been investigated for functions related to cancer biology, whereas others have not⁸⁻¹⁰. Especially in ovarian cancer, the function of most miRNAs are unknown. From this standpoint, we believe that this study can provide useful information for further functional investigations.

References;

1. Ge, Q., *et al.* miRNA in plasma exosome is stable under different storage conditions. *Molecules* **19**, 1568-1575 (2014).
2. Sourvinou, I.S., Markou, A. & Lianidou, E.S. Quantification of circulating miRNAs in plasma: effect of preanalytical and analytical parameters on their isolation and stability. *J Mol Diagn* **15**, 827-834 (2013).
3. Langhe, R., *et al.* A novel serum microRNA panel to discriminate benign from malignant ovarian disease. *Cancer Lett* **356**, 628-636 (2015).
4. Pendlebury, A., *et al.* The circulating microRNA-200 family in whole blood are potential biomarkers for high-grade serous epithelial ovarian cancer. *Biomed Rep* **6**, 319-322 (2017).
5. Shapira, I., *et al.* Circulating biomarkers for detection of ovarian cancer and predicting cancer outcomes. *Br J Cancer* **110**, 976-983 (2014).
6. Vaughan, S., *et al.* Rethinking ovarian cancer: recommendations for improving outcomes. *Nat Rev Cancer* **11**, 719-725 (2011).
7. Witwer, K.W., *et al.* Standardization of sample collection, isolation and analysis methods in extracellular vesicle research. *J Extracell Vesicles* **2**(2013).
8. Santos, J.C., Ribeiro, M.L., Sarian, L.O., Ortega, M.M. & Derchain, S.F. Exosomes-mediate microRNAs transfer in breast cancer chemoresistance regulation. *Am J Cancer Res* **6**, 2129-2139 (2016).
9. Sun, J.Y., *et al.* MicroRNA-320a suppresses human colon cancer cell proliferation by directly targeting beta-catenin. *Biochem Biophys Res Commun* **420**, 787-792 (2012).
10. Cai, H., *et al.* Epigenetic inhibition of miR-663b by long non-coding RNA HOTAIR promotes pancreatic cancer cell proliferation via up-regulation of insulin-like growth factor 2. *Oncotarget* **7**, 86857-86870 (2016).

REVIEWERS' COMMENTS:

Reviewer #1 (Remarks to the Author):

The authors have revised the manuscript in response to comments from all three reviewers.

This reviewer made several specific comments

1. The purpose of the analysis: to act as a population-based primary screening tool (cancer vs no pathology) OR to increase diagnostic accuracy in those suspected of having possible ovarian cancer (cancer vs benign pathology). The performance characteristics for such as a primary screening tool are exacting and the downstream validation will require evaluation in many thousands of women (UKCTOCS recruited >200,000 women). However, none of the models presented by the authors was able to discriminate between ovarian cancer and non-malignant ovarian tumours (benign or borderline), so the authors may be correct that this is not going the future role of miRNA assessment. The authors have, partially at least, addressed this point.
2. Validation of the miRNA. This reviewer felt strongly that validation of the miRNA identified in arrays was important. The authors have undertaken qRT-PCR analysis of 10 miRNA per model (Supplementary Figure 2), choosing those with the highest individual AUC, and then correlating qRT-PCR and microarray data (Supplementary Figure 3). Unsurprisingly, not all results validated and correlated. This validation is important because it is very unlikely that any future miRNA assay will be array-based. However, the authors have addressed my point.
3. SOP for serum collection. I expressed concern that samples were stored at 4°C for one week prior to analysis. The authors have addressed this point as well as they can in the discussion and note that they are undertaking prospective sample collection.
4. Cell lines. The shortcomings of ovarian cancer cell lines are now well-described in the literature. I remain concerned that SKOV3 and A2780 are included in the original cell lines. However, the authors raise the point that the lines that chose cover a range of ovarian cancer histological types and thus may allow identification of miRNA specific to clear cell and endometrioid tumours.
5. The lack of difference between miRNA from different ovarian cancer types. The results are intriguing and suggest that the miRNA released from all types of ovarian cancer are largely similar. This is particularly interesting given that model 2 was (at least partially) able to distinguish ovarian cancers from other solid malignancies.

Overall, the authors have made valid attempts to address the reviewer's comments, although I still have concerns about the downstream applicability of these data in a screening strategy.

Minor point

Reviewer #2 (Remarks to the Author):

The authors have addressed the issues raised.

Reviewer #3 (Remarks to the Author):

While some of the inherent flaws of the study persist, I do appreciate the authors' interval efforts to provide further data and reframe the text in a more tempered fashion. No additional questions exist.

August 16th, 2018

RE: Integrated extracellular microRNA profiling for ovarian cancer screening

Manuscript # NCOMMS-17-32611

Responses to reviewer's comments

Reviewer #1

We really appreciate your helpful discussion, which lead to improve our manuscript. All of the points you raised have been addressed as followed.

Comment_1: The purpose of the analysis: to act as a population-based primary screening tool (cancer vs no pathology) OR to increase diagnostic accuracy in those suspected of having possible ovarian cancer (cancer vs benign pathology). The performance characteristics for such as a primary screening tool are exacting and the downstream validation will require evaluation in many thousands of women (UKCTOCS recruited >200,000 women). However, none of the models presented by the authors was able to discriminate between ovarian cancer and non-malignant ovarian tumours (benign or borderline), so the authors may be correct that this is not going the future role of miRNA assessment. The authors have, partially at least, addressed this point.

Response: We really appreciate that you understand our purposes this study. To distinguish ovarian cancer from non-malignant ovarian tumours, we have to keep making an effort to generate models by another approach, and we would like to consider it as future works.

Comment_2: Validation of the miRNA. This reviewer felt strongly that validation of the miRNA identified in arrays was important. The authors have undertaken qRT-PCR analysis of 10 miRNA per model (Supplementary Figure 2), choosing those with the highest individual AUC, and then correlating qRT-PCR and microarray data (Supplementary Figure 3). Unsurprisingly, not all results validated and correlated. This validation is important because it is very unlikely that any future miRNA assay will be array-based. However, the authors have addressed my point.

Response: We totally agreed with your suggestion and the process of qRT-PCR validation was really worth to do. We really appreciate that you consider that we could address the points.

Comment_3: SOP for serum collection. I expressed concern that samples were stored at 4°C for one week prior to analysis. The authors have addressed this point as well as they can in the discussion and note that they are undertaking prospective sample collection.

Response: The parts you pointed out are still limitations of our work. However, we would like to answer these concerns in future works.

Comment_4: Cell lines. The shortcomings of ovarian cancer cell lines are now well-described in the literature. I remain concerned that SKOV3 and A2780 are included in the original cell lines. However, the authors raise the point that the lines that chose cover a range of ovarian cancer histological types and thus may allow identification of miRNA specific to clear cell and endometrioid tumours.

Response: We really appreciate that you understand the rational for selecting cell lines.

Comment_5: The lack of difference between miRNA from different ovarian cancer types. The results are intriguing and suggest that the miRNA released from all types of ovarian cancer are largely similar. This is particularly interesting given that model 2 was (at least partially) able to distinguish ovarian cancers from other solid malignancies. Overall, the authors have made valid attempts to address the reviewer's comments, although I still have concerns about the downstream applicability of these data in a screening strategy.

Response: As you mentioned, we tried to distinguish ovarian cancers from other solid malignancies but it did not work with perfect quality. In this study, we focused on whole circulating miRNAs. If we could analyze some specific types of circulating miRNAs, such as miRNAs encapsulated in some specific marker-positive exosomes, prediction performance might improve. We would like to make these approach as future work.

We believe that we have successfully addressed all of the issues raised by the reviewers and would like to thank all reviewers for helpful discussion.

Sincerely,

Takahiro Ochiya, Ph.D.

Chief,

Division of Molecular and Cellular Medicine,

National Cancer Center Research Institute,

5-1-1, Tsukiji, Chuo-ku, Tokyo 104-0045 Japan

[Tel:+81-3-3547-5276](tel:+81-3-3547-5276)

Fax:+81-3-5565-0727

tochiya@ncc.go.jp